# Persistent and Dose-Dependent Neural and Metabolic Gene Expression Changes Induced by Transient Citalopram Exposure in Zebrafish Embryos

**DOI:** 10.3390/ijms262311288

**Published:** 2025-11-22

**Authors:** Ryan J. North, Gwendolyn Cooper, Lucas Mears, Brian Bothner, Mensur Dlakić, Christa S. Merzdorf

**Affiliations:** 1Department of Microbiology and Cell Biology, Montana State University, Bozeman, MT 59717, USA; ryannorth@montana.edu (R.J.N.); lucashmears@gmail.com (L.M.);; 2Department of Chemistry and Biochemistry, Montana State University, Bozeman, MT 59717, USA; gwendolyncooper@montana.edu (G.C.); bbothner@montana.edu (B.B.)

**Keywords:** citalopram, zebrafish, neural development, RNA-seq, hormetic dose–response, metabolic regulation, SSRI, developmental toxicology

## Abstract

Citalopram, a common selective serotonin reuptake inhibitor (SSRI), has been increasingly detected in aquatic environments due to ineffective removal and improper disposal. Although developmental exposure to SSRIs is linked to neurotoxicity, little is known about the persistence of gene expression alterations following limited exposure periods. Zebrafish embryos were exposed from 2 to 24 h post-fertilization (hpf) at concentrations of citalopram hydrobromide spanning surface water to therapeutic serum levels (0.03, 0.9, 50, and 250 μg/L), followed by removal of the citalopram and development until 48 hpf. Whole-embryo RNA sequencing at 48 hpf revealed a non-linear dose–response wherein the lowest dose resulted in the induction of the highest number of differentially expressed genes (DEGs). Gene set enrichment analyses (GSEA) and overrepresentation analyses (ORAs) showed that 0.03 μg/L citalopram caused upregulation of metabolic and developmental pathway genes, but suppressed synaptic membrane genes, whereas 0.9 μg/L resulted in strong downregulation of key neurotransmitter receptors. At 50 μg/L, genes linked to oxidative stress (glutathione metabolism and ferroptosis) were upregulated, and at 250 μg/L, stress and apoptotic processes were increased, while glutamate receptor genes were repressed. All four citalopram doses suggested synaptic and neurotransmitter alterations, implying that persistent neurodevelopmental impacts resulted from a limited early window of exposure. These data highlight that transient, low-level SSRI exposures shape long-term embryonic gene expression.

## 1. Introduction

Citalopram is a selective serotonin reuptake inhibitor (SSRI) commonly prescribed to treat major depressive disorders as well as other mental and behavioral disorders [1]. With citalopram and escitalopram (the purified S enantiomer) growing rapidly in use, unintended human and animal exposures have become more common [2,3,4,5,6,7]. Besides direct input into wastewater (ex: flushing of unused medication), 12–23% of ingested citalopram is excreted unmetabolized by the human body [8]. As a result, citalopram and its metabolites have been detected at concentrations ranging from approximately 0.1 to 1 μg/L in wastewater effluent and 1 to 100 ng/L in surface waters downstream of treatment plants worldwide [9,10,11,12]. These compounds have also been found in aquatic animals such as fish and mollusks in both fresh and saltwater ecosystems, suggesting a broad environmental scope [7,13]. The presence of these pharmaceuticals in water supplies raises concern for both environmental and public health.

Selective serotonin reuptake inhibitors, such as citalopram, inhibit the serotonin transporter (SERT), preventing the reuptake of serotonin (5-hydroxytryptamine, 5-HT) into presynaptic neurons and thereby prolonging serotonergic signaling [1,14]. While SSRIs act primarily on the central nervous system (CNS), approximately 90% of the body’s serotonin is synthesized in the gastrointestinal tract [15], where L-tryptophan is hydroxylated by tryptophan hydroxylase 1 (TPH1) and then decarboxylated for form 5-HT [16,17]. Peripheral serotonin cannot cross the blood–brain barrier (BBB) [18], but can indirectly influence brain function through vagal nerve activation, modulation of immune responses, and alterations in tryptophan metabolism [19,20,21]. Interestingly, L-tryptophan can cross the BBB and serves as a prerequisite molecule for central 5-HT synthesis via neuronal tryptophan hydroxylase 2 (TPH2) [22]. Within presynaptic neurons, 5-HT is loaded into synaptic vesicles by the vesicular monoamine transporter 2 (VMAT2/SLC18A2), released into the synaptic cleft via calcium-dependent exocytosis, and degraded post-release by monoamine oxidase (MAO) into 5-hydroxyindoleacetic acid (5-HIAA) [23,24]. Serotonin release is further regulated by presynaptic autoreceptors (ex: HTR1A, HTR1B, HTR1D), which modulate neurotransmitter release through feedback inhibition [24].

On the postsynaptic side, 5-HT binds G protein-coupled receptor (GPCR) subtypes HTR1, 2, 4–7 and the ligand-gated ion channel HTR3 [25,26,27]. HTR1/HTR5 typically reduce intracellular cAMP via Gi/o, while HTR4/HTR6/HTR7 raise cAMP through coupling with Gs proteins [25]. HTR2 activates phospholipase C to increase IP3 and DAG; and HTR3 governs rapid depolarization as a Na+/K+ ion channels [24,27]. These cascades feed into downstream pathways, including cAMP/protein kinase A (PKA), mitogen-activated protein kinase (MAPK), Wnt, and NF-κB, that modulate neuronal plasticity, survival, and synaptic connectivity [27,28]. Through these diverse signaling mechanisms, serotonin exerts broad control over nervous system formation, function, and adaptability.

Serotonin plays a critical role during early embryogenesis, influencing processes such as cell proliferation, migration, and axon guidance [23,29]. During early development in mammals, serotonin functions as a growth factor, regulating neural progenitor proliferation, neurite extension, and synaptogenesis, partly through the modulation of Wnt, MAPK, and SHH signaling pathways [30,31,32]. Serotonin signaling is central to the development of the hypothalamic–pituitary–adrenal axis and the locus coeruleus-norepinephrine system, which shape stress response and autonomic regulation [33,34]. It also contributes to cardiac morphogenesis, as evidenced by embryonic lethality in HTR2B knockout mice due to severe heart defects [35]. SSRIs such as paroxetine and fluoxetine can cross the placental barrier and have been associated with a modest increase in cardiac defects such as atrial septal defects [1,36]. Overall, disruptions to these developmental signaling networks are associated with congenital abnormalities and long-term behavioral outcomes [37,38].

Zebrafish have been used extensively as a model system for studying the developmental consequences of toxicant exposure [39]. The first 24 h of zebrafish development are highly comparable to the first month of human gestation. Multiple developmental processes are shared during the next 24 h of zebrafish development and the second month of gestation in human [40,41,42], after which the two organisms diverge. Since early processes of embryogenesis are largely conserved between humans and zebrafish, so too are serotonergic and neurotransmission processes, thereby offering insights into human development [39,40,41,42]. Given this utility, zebrafish are an important model for assessing the developmental impacts of pharmaceuticals, including SSRIs.

Previous studies in zebrafish predominantly characterized the neurotoxic effects of developmental citalopram exposure [43,44]. In that work, continuous exposure to citalopram was maintained up to the time of harvest, with time points extending beyond 48 hpf. Exposure to 10 μg/L increased mortality by 24 hpf and produced dose-dependent reductions in hatching rate and heart rate. By 48 hpf, concentrations of 1 and 10 μg/L induced yolk sac and pericardial edema as well as scoliosis. At the gene expression level, 96 h exposures at 1 and 10 μg/L altered oxidative stress markers, including superoxide dismutase, catalase, and glutathione-S-transferase- and significantly decreased acetylcholine esterase activity, while also increasing apoptosis, lipogenesis, and bone mineralization. Longer-term studies extending from embryonic stages to adulthood (0–150 days) across 0.1 to 100 μg/L reported altered locomotor behavior linked to reduced numbers of glutamatergic spinal motor neurons, with the highest dose also diminishing dopaminergic neuron counts [45]. In adult zebrafish, a two-week exposure of 4 and 40 μg/L resulted in decreased expression of gnrh3, gonadotropins, and serotonergic genes, suggesting impaired spermatogenesis [46]. Furthermore, exposures from 0 to 7 dpf over a broad concentration range (0.1–1000 μg/L) corroborated the dose-dependent bradycardia, increased reactive oxygen species, and cardiac gene expression alterations, with RNAseq analyses revealing perturbations in pathways related to RNA polymerase activity, regeneration, glutamate receptors, NADH ubiquinone oxidoreductase, myosin, and troponin [47]. Similar gene expression changes following a 6-day exposure to 100 μg/L fluoxetine further underscore the shared molecular targets among selective serotonin reuptake inhibitors and demonstrate significant effects on endocrine function [48]. However, the continuous exposure regimens used in these studies make it difficult to distinguish between the transient and persistent effects of citalopram.

Although previous studies have begun to examine the developmental effects of SSRIs, we uniquely allowed our embryos a citalopram-free period prior to harvest in order to eliminate gene expression changes resulting from acute citalopram exposure. Further, we tested a lower, more environmentally relevant concentration than previous studies. Finally, we focused on the very early effects of developmental citalopram exposure. By analyzing embryos at 48 hpf, we captured changes potentially relevant to human development. Further, we analyze an early time point that has not been well-studied in the existing literature. We employed whole-embryos RNA sequencing to identify persistent transcriptomic alterations following discontinued early development exposure to citalopram across a range of concentrations that reflected both environmental and therapeutic levels.

## 2. Results

### 2.1. Transcriptomic Shifts in Citalopram Conditions

Zebrafish embryos were exposed from 2 to 24 hpf to concentrations of citalopram that included 0.03 ug/L (environmental surface water), 0.9 ug/L (wastewater treatment plant effluent), 50 ug/L (therapeutic human serum concentration), and 250 ug/L (super-therapeutic serum concentration) [49,50]. After 24 h of treatment-free development, the embryos were harvested at 48 hpf for RNA extraction. Following sequencing, we conducted exploratory analyses of our RNAseq data to determine global gene expression variance across treatment groups and identify potential dose-dependent trends. To assess intra-and-inter condition variance, we constructed principal component analysis (PCA) plots. PCA of Salmon-derived normalized transcript per million (TPM) counts for control and citalopram-treated conditions (0.03, 0.9, 50, and 250 µg/L) showed considerable overlap among treatment groups vs. control, indicating minimal overall intergroup variance (Figure 1A–D). Despite the overlap of confidence intervals, a significant amount of variation was captured by PC1 and PC2. For each citalopram condition versus control the following PC1/PC2 percentages of variance were demonstrated: 32.21%/14.9% (0.03 µg/L), 24.7%/15.21% (0.9 µg/L), 19.38%/18.05% (50 µg/L), and 27.18%/17.68% (250 µg/L) (Figure 1A–D). In all cases, the citalopram replicates (*n* = 5 for each condition) remained within the control group’s 95% confidence interval, suggesting minimal differentiation by PCA.

In contrast, partial least squares-discriminant analysis (PLS-DA), a supervised technique that is particularly useful for identifying biomarkers and distinguishing between physiological states, demonstrated clear separation in two variates (Figure 2A–D). The respective X axis variates (X-variate 1) and Y-axis variates (X-variate 2) explained variance values were 30%/10%, 8%/18%, 13%/13%, and 9%/22% for the 0.03, 0.9, 50, and 250 μg/L conditions, respectively (Figure 2A–D). Thus, while PCA was insufficient to discriminate treatments from control, PLS-DA identified separable variance, with each treatment condition showing a combined explained variance above 25%. These data indicated that while the majority of transcript levels were similar between control vs. treatment groups (Figure 2A–D), there was enough unique variance between each group that they could be separated by PLS-DA techniques (Figure 2A–D).

### 2.2. Differentially Expressed Transcripts of Citalopram Exposure

To determine significant gene expression changes induced by each citalopram concentration we derived differentially expressed transcripts (DETs) from our mapping and counts data. Salmon-derived raw counts were analyzed in DESeq2 to identify DETs at the four citalopram concentrations (0.03, 0.9, 50, and 250 μg/L) compared to controls (Figure 3A–D). Transcripts were considered DETs if they had an adjusted *p*-value (p_adj_) ≤ 0.05 (determined by DESeq2) and an absolute log_2_ fold change (|log_2_FC|) > 0.6. Number of DETs for each condition (*n* = 5) relative to control: 0.03 ng/mL yielded 209 DETs (189 upregulated, 20 downregulated), 0.9 ng/mL yielded 13 DETs (4 upregulated, 9 downregulated), 50 ng/mL yielded 11 DETs (6 upregulated, 5 downregulated), and 250 ng/mL yielded 28 DETs (23 upregulated, 5 downregulated) (Figure 3E). These results indicated a non-linear dose response, with the lowest dose (0.03 μg/L) producing the greatest number of DETs, intermediate doses (0.9 and 50 μg/L) yielding the fewest, and the highest dose (250 μg/L) producing an intermediate response. The directionality of expression also varied with dose: the ratio of upregulated to downregulated DETs was approximately 9:1 at 0.03 μg/L, 0.4:1 at 0.9 μg/L, 1.2:1 at 50 μg/L, and 4.6:1 at 250 μg/L. These ratios reflect a dose-dependent shift in directionality of differential expression, with upregulation dominating at low and high doses, and downregulation having a stronger influence at intermediate doses. Correspondingly, the mean log_2_FC values for upregulated transcripts ranged from 15.25 to 19.41, whereas downregulated transcripts ranged from −9.78 to −22.67 (Figure 3A–D). This indicated that both up- and downregulated transcripts exhibited large changes in magnitude across doses. Collectively, these data suggested that certain genes were particularly sensitive to low-dose citalopram and that the magnitude and direction of changes in gene expression varied according to citalopram concentration.

### 2.3. Pathway Analysis: Over-Representation Analysis and STRING

To determine biological implications of citalopram exposure, we examined pathways, functional categories, and transcriptional regulators associated with the differentially expressed genes (DEGs). At 0.03 μg/L citalopram, no significantly enriched KEGG or GO terms were detected under stringent criteria (*p* < 0.05, FDR < 0.05) (Appendix A). However, applying an exploratory threshold (*p* < 0.05, FDR < 0.20) revealed potential enrichments for the KEGG term: Ribosome, and for multiple GO:BP terms, including: embryo development ending in birth or egg hatching, tube formation, head development, cell population proliferation, renal system development, regeneration, cellular modified amino acid metabolic process, ectodermal placode development, ear development, and amoeboid-type cell migration (Appendix A). No GO:CC or GO:MF terms met the exploratory threshold at the 0.03 μg/L dose. Subsequent analyses focused on this dose, as no other concentrations produced significant KEGG or GO term enrichments under either stringent or exploratory thresholds.

Transcription factor (TF) enrichment analysis revealed that DEGs at 0.03 μg/L shared motifs associated with factors such as YCATTAA_Unknown, AAAYRNCTG_Unknown, FOXC1, HNF1_Q6, HOXA4_Q2, MAF_Q6, TGTTGY_HNF3_Q6, YY1_01_EGR1_01, and CDC5_01 were detected. In contrast, the 0.9, 50, and 250 ng/mL treatments did not yield any significant shared TF binding motifs. This suggests that the lowest concentration of citalopram elicited a more coordinated regulatory impact on gene expression than higher concentrations.

In order to investigate potential functional interactions, all differentially expressed genes (DEGs) identified at each citalopram concentration were submitted to STRING in three separate sets: total DEGs, upregulated, and downregulated. The three sets were used to discern potentially separate biological processes that are driven by either up- or by downregulation. At 0.03 μg/L, the downregulated DEG list did not yield meaningful clusters, whereas the upregulated DEGs formed a prominent interaction network centering on ribosomal function. The total set of DEGs from the 0.03 μg/L condition produced a similar cluster to the upregulated set driven by the same DEGs. This network showed enrichment for KEGG/GO terms related to eukaryotic translation initiation, translational factor activity/RNA binding, cytoplasmic ribosomal proteins, and the “Ribosome” pathway, alongside biological processes such as positive regulation of transcription, animal organ development, regulation of primary metabolic processes, and regulation of cell population proliferation (Appendix A). At 0.9 μg/L, no significant clusters were identified, but a small network was observed linking two downregulated genes (*eif1axb*, which codes for a eukaryotic translation initiation factor and *rpl15a*, which codes for a subunit of the large ribosomal subunit) with one upregulated gene (*myb* which encodes a transcriptional regulator essential for hematopoiesis). The 50 μg/L dose yielded no notable clusters or connections among the DEGs, while at 250 μg/L two short interaction chains emerged among the upregulated genes: *Rpl9* and *Rpl30*, both encoding ribosomal proteins, and *Pnrc2* which encodes a nuclear receptor coactivator protein, as well as *rasgrp4* (encoding RAS guanyl releasing protein 4) and *cacna1ba* (encoding the N-type voltage-dependent calcium channel alpha 1B subunit A), despite the absence of significant larger clusters.

### 2.4. Gene-Set Enrichment Analysis Reveals Altered Pathways

To capture more global coordinated changes in biological pathways beyond what discrete thresholds reveal, we performed gene set enrichment analysis (GSEA). Over-representation analysis (ORA) often employs discrete cutoffs on log_2_FC and p_adj_, which, while valid, can overlook large-scale or subtle multi-gene shifts [51,52,53]. Balancing ORA and GSEA thus provides a method to detect both highly significant genes and more diffuse yet coordinated pathway alterations [52].

At 0.03 μg/L, multiple KEGG pathways were enriched, including Glycine, Serine, and Threonine metabolism (NES = 1.83), Adherens junction (1.82), TGF-beta signaling (1.72), and Oxidative phosphorylation (1.71) (Appendix A). GO enrichments at this dose encompassed ectodermal placode development (1.86) and regulation of membrane potential (−2.03) in BP, collagen trimer (2.03) and synaptic membrane (−1.81) in CC, and structural constituent of eye lens (2.29) and neurotransmitter receptor activity (−2.09) in MF (Appendix A). Upregulated pathways predominated over downregulated terms. Transcription factor target analysis indicated enrichment for EGFR (NES = 1.93), SQSTM1 (1.91), ZMYND11 (1.90), HOXC13 (1.87), and RUVBL2 (1.76) target genes, as well as negative enrichments for BAHD1 (−2.56), MYF6 (−2.52), SMCHD1 (−2.40), ZNF586 (−2.33), and ZNF418 (−2.29) (Table 1).

At 0.9 μg/L, the analysis revealed significant GO:CC terms such as synaptic membrane (−1.95), post synapse (−1.90), and axon (−1.86). In addition, structural constituent of eye lens (2.17) and organophosphate ester transmembrane transporter activity (2.00) emerged as enriched GO:MF terms (Appendix A). Downregulated pathways outnumbered the upregulated pathways and STN1 (1.86) was the only significantly enriched transcription factor target (Table 1).

At 50 μg/L, numerous KEGG terms were enriched, including arachidonic acid metabolism (2.09), glutathione metabolism (2.08), ferroptosis (2.01), glycolipid metabolism (1.98), biosynthesis of cofactors (1.91), and tight junction (1.91), whereas cardiac muscle contraction (−2.00), ATP-dependent chromatin remodeling (−1.84), neuroactive ligand-receptor interaction (−1.82), and calcium signaling pathway (−1.64) were negatively enriched (Appendix A). GO:BP categories spanned skin development (2.20), pigment metabolic process (2.13), response to toxic substance (2.11), and synaptic signaling (−2.78). GO:CC enrichments included collagen trimer (1.89) and axon (−2.71), while GO:MF highlighted cysteine-type endopeptidase regulator activity in apoptosis (2.32) and glutamate receptor activity (−2.45) (Appendix A). The upregulated-to-downregulated ratio approached equal at 0.94:1. Transcription factor target analysis revealed significant enrichment for PSMB5 (1.65) and MED25 (1.62) target genes, with negative enrichment for NRSF_01 (−2.57), FOXM1_01 (−2.22), OCT1_02/04/06 (−2.19, −2.21, and −2.20, respectively), AHRARNT_01 (−2.20), CDP_02 (−2.19), and HNF6_Q6 (−2.17) (Table 1).

Finally, at 250 μg/L, significantly enriched GO:BP terms included cellular response to external stimulus (1.91) and synaptic signaling (−2.14) (Appendix A). GO:CC and GO:MF analysis underscored enrichment for the basal part of the cell (1.68) and downregulation in neurotransmitter receptor activity (−2.28) and glutamate receptor activity (−2.15) (Appendix A). Overall, downregulated terms predominated with an (Up/Down) ratio of 0.15:1. Transcription factor target analysis revealed significant enrichment for HSF4 (2.05) and ZA_UNIPROT_Q9UM89 (1.64), with negative enrichments for MSX2 (−2.25), MYF6 (−2.22), SMCH1 (−2.11), ZNF510 (−2.08), ZNF202 (−2.04), and SOX11 (−2.04) (Table 1).

While ORA identified few enriched pathways at both strict and relaxed thresholds, GSEA uncovered a broader spectrum of coordinated pathway changes across all doses. This approach revealed dose-specific and shared alterations, particularly in synaptic, metabolic, and developmental processes, that were not apparent with threshold-based analyses alone. This underscores the value of GSEA in detecting subtle transcriptomic shifts.

### 2.5. Citalopram Did Not Affect Cell-Type Populations

Among 135 cell types that occur at 48 hpf in the Atlas 1.0 single cell dataset, our gene expression profiles discerned 45 unique cell types, which appeared in all conditions. Relative to control, no cell type showed significant changes in its proportion value in response to any dose of citalopram (Appendix A). Performing the analysis with raw unadjusted *p* values suggested that the dose of 0.03 μg/L may have affected ciliated spinal cord neurons (*p* = 0.018, p_adj_ = 0.279), lens cells (*p* = 0.031, p_ad_j = 0.279), neutrophils (*p* = 0.039, p_adj_ = 0.279), mid-hindbrain boundary neurons (*p* = 0.048, p_adj_ = 0.279), and fin bud cells (*p* = 0.048, p_adj_= 0.279); 0.9 μg/L may have affected hair cell neurons (*p* = 0.049, p_adj_ = 0.279); 50 μg/L had no raw *p*-values less than 0.05; and 250 μg/L may have affected neural crest cells (*p* = 0.035, p_adj_ = 0.279) (Appendix A).

## 3. Discussion

Zebrafish are a powerful vertebrate model for studying the developmental effects of pharmaceuticals such as citalopram due to their ease of use and genetic and molecular similarity to humans. In a discontinued exposure regimen, zebrafish embryos were exposed to citalopram hydrobromide from 2 to 24 hpf, followed by a 24 h drug-free period and harvest at 48 hpf (Figure 1A). The tested concentrations (0.03, 0.9, 50, and 250 μg/L citalopram) were selected to represent environmentally relevant surface water levels, wastewater effluent levels, therapeutic human serum concentrations, and super-therapeutic serum concentrations, respectively. Transient citalopram exposure during early zebrafish development led to persistent gene expression alterations detectable well after cessation of exposure. Thus, our data suggest that modulation of serotonin signaling during the first 24 h of development can alter neurodevelopmental pathways in zebrafish.

### 3.1. Effects on Differential Expression Are Persistent and Non-Linear

RNA-seq analysis revealed a non-linear dose–response in which the lowest dose of citalopram (0.03 μg/L) produced the highest number of differentially expressed genes (DEGs), the intermediate doses (0.9 and 50 μg/L) yielded the fewest, and the highest dose (250 μg/L) led to a roughly twofold increase in DEGs relative to the 50 μg/L dose (Figure 3E). This pattern is indicative of a hormetic response. Similar biphasic effects have been reported in response to opiates and lead [54,55] and other citalopram studies in response to different doses, as well as acute and chronic exposure [49,50]. In mice, acute administration of a high dose of citalopram produced anxiogenic behavior, whereas repeated administration of lower doses over 24 h produced anxiolytic behavior, demonstrating that different dose and administration regimens can elicit nearly opposite responses [49]. In rats, chronic citalopram treatments led to dose-dependent changes in serotonin signaling in a non-linear fashion [56,57]. Additional studies in developmental models (hESCs, rats, and mice) indicate that the effects of serotonin signaling are both dose- and time-dependent [30,58,59,60].

One explanation for the non-linear effects may involve the activity of serotonin autoreceptors, which are specialized receptors on the presynaptic neuron that detect neurotransmitter concentrations and initiate a negative feedback, preventing further serotonin release [61]. During early zebrafish development, corresponding to the exposure window in this study (0–24 hpf), serotonin receptor coding genes, including autoreceptors, are expressed [62]. At low serotonin concentrations, incomplete activation and subsequent lack of desensitization of 5-HT1A autoreceptors (coded by the hrt1aa and hrt1ab genes in zebrafish) may allow a stronger downstream transcriptional response, whereas higher doses may trigger strong autoreceptor activation and compensatory feedback mechanisms [63]. This means that at low serotonin doses, autoreceptors are only mildly activated, so synaptic serotonin remains elevated; at higher doses, autoreceptors are strongly activated, leading to reduced serotonin signaling. The dose-dependent activation of autoreceptors is made more complex by desensitization in which receptors become less responsive to serotonin in a dose-dependent manner [64,65]. Chronic high-dose exposure to citalopram may lead to autoreceptor desensitization, restoring serotonin release [66], which may explain the increase in DEGs at the highest dose.

These data suggested that the non-linear effects of citalopram arise from dose-dependent regulation of gene expression rather than a uniform transcriptional response. To evaluate whether these transcriptional differences might instead reflect changes in cell-type abundance, we performed single-cell deconvolution. The absence of significant compositional shifts in cell types across doses indicated that early serotonergic perturbation influenced transcriptional programs within stable developmental cell populations. Thus, the observed differences likely represent persistent regulatory reprogramming rather than altered cell-lineage effects, though subtle shifts in rare or transient populations may emerge with larger sample sizes. Understanding these Dose–Response relationships is essential for evaluating the real-world developmental impacts of low-level citalopram exposure.

### 3.2. Dose-Specific and Shared Pathway Disruptions

Our analysis demonstrated that transient citalopram exposure during development induced persistent transcriptomic alterations at 48 hpf. While stringent over-representation analysis (ORA), which tests whether predefined pathways are statistically overrepresented in a filtered gene list, detected few enriched terms at any dose, gene set enrichment analysis (GSEA), which evaluates whether predefined pathways are consistently shifted towards up- or downregulation across all ranked genes, revealed significant KEGG and GO term enrichment. This discrepancy illustrates that threshold-based ORA may miss coordinated, subtle changes that GSEA can capture.

GSEA showed that across all doses, synaptic and neuronal signaling pathways were downregulated, indicating a shared impairment in synapse function, neurotransmission, and plasticity (Appendix A). Even at 0.03 μg/L citalopram, genes associated with neurotransmitter receptor activity were suppressed, despite the broader trend towards transcriptomic upregulation. At 0.9 μg/L and 50 μg/L citalopram, this suppression was more pronounced, encompassing terms such as synaptic membrane, axonal components, and postsynaptic compartments (Appendix A). At each dose, these enrichments were driven by a relatively small subset of strongly downregulated genes with central roles in these pathways such as *cdh2* (neuronal cadherin) [67], *grin1a* and *grin2ca* (subtypes of the glutamate ionotropic receptor) [68]. This means that even modest numbers of DEGs can yield significant pathway enrichment if those genes are functionally essential or highly connected. Additionally, GSEA can detect pathway-level shifts even when only part of the gene set changes due to consideration of coordinated direction trends across all ranked genes rather than just significant ones.

At 0.03 μg/L, pathway enrichment indicated activation of metabolic and developmental processes, including oxidative phosphorylation, glycine/serine/threonine metabolism, adherens junction formation, and ectodermal placode development (Appendix A). This overlap between GSEA and ORA, such as in ectodermal placode development, adds robustness to these findings. Notable DEGs further linking these pathways to neuronal differentiation, cell adhesion, and cytoskeletal reorganization include the LINGO1, AUTS2, foxp1a, tfap2a, and map1ab genes (Appendix A).

At 50 and 250 μg/L, pathway enrichment shifted towards metabolic stress, redox imbalance, and apoptosis (Appendix A). Upregulation of pathways such as arachidonic acid metabolism, glutathione metabolism, ferroptosis, and tight junction formation indicates increasing cellular stress (Appendix A). These findings are consistent with prior studies showing that SSRIs disrupt mitochondrial function and redox balance [69,70,71], and that ferroptosis (iron-dependent cell death) may be linked to depressive phenotypes and is modulated by SSRIs [72,73]. Together these data suggest that, while lower doses primarily affect developmental and synaptic pathways, concentrations above 50 μg/L induce a robust stress response and compensatory mechanisms.

These findings suggest that environmentally relevant, very low-level exposures to SSRIs can alter key neurodevelopmental pathways. Such alterations may have long-term implications for neural circuit formation, function, and behavior.

### 3.3. Transcription Factor Shifts Align with Differential Expression

Transcription factor (TF) enrichment patterns reflected the dose-dependent shifts observed in our gene expression and pathway analysis. Across doses, changes in postulated upstream TFs were consistent with the suppression of neuronal signaling and activation of metabolic or stress-related gene expression. At the lowest dose (0.03 μg/L citalopram), this analysis pointed to TFs that act as developmental regulators (Table 1), many of which control the expression of genes involved in neuronal differentiation [74,75,76], adhesion [67], and mitochondrial function [77,78,79]. Negative enrichment for neuronal repressors (Table 1) further suggested promotion of neural development [80,81,82]. Intermediate doses (0.9 and 50 μg/L citalopram) showed fewer TF enrichments but included stress/cell cycle regulators [83,84], accompanied by negative enrichment of neuronal repressors [85,86,87] (Table 1), potentially indicating a targeted impact on neurotransmission. The highest dose (250 μg/L citalopram) had TF profiles which shifted toward stress and survival responses, including enrichment for HSF4 and zinc-finger TFs, alongside negative enrichment of developmental regulators [80,88,89,90,91] (Table 1). This pattern was elicited by upregulation of metabolic redox genes [92,93] and downregulation of neuronal components. Although no single TF target was shared across all doses, several neuronal genes (*cdh2*, *grin1a*, *slc17a6b*) recurred in multiple citalopram exposures.

These findings deepen our understanding of the lasting effects of citalopram exposure during development as a transcription factor-driven process. At low concentrations (0.03 μg/L), we see a generally pro-development pattern accompanied by some targeted neuronal TF repression; intermediate doses (0.9 and 50 μg/L) hold important gene sets targeted by neuronal repressors and stress/cell-cycle mediators; and at the high doses we see a shift towards heat shock and pro-survival transcription factors with downregulation of neuronal differentiation factors.

### 3.4. Synaptic and Neuronal Function

In all exposure conditions, gene set enrichment analysis (GSEA) consistently revealed significant and persistent downregulation of genes involved in synaptic and neuronal function well past the citalopram exposure window (Appendix A). Because GSEA evaluates ranked gene lists in their entirety, these pathway enrichments may reflect coordinated shifts across many genes within a pathway rather than changes in a large number of individual DEGs. This persistent, citalopram-induced downregulation suggested that early developmental perturbations in serotonergic signaling led to long lasting impairments in neuronal connectivity and communication. Previous studies also suggest this interpretation. For instance, neonatal citalopram exposure in rats reduces serotonin synthesis via decreased tryptophan hydroxylase activity, and altered serotonin transporter expression persists [94]. These molecular changes correlate with behavioral alterations in rodents, which include increased locomotor activity and decreased sexual behavior [94], and mirror depressive symptoms that can be reversed by subsequent antidepressant treatment [95].

The term “synaptic membrane” was reported in all citalopram doses except 50 μg/L, with the 0.03 μg/L dose showing the highest enrichment. Within this term, critical genes include neural cadherin (*cdh2*), a calcium-dependent cell adhesion molecule essential for neural cell–cell adhesion that is robustly associated with neurodevelopment and neurodegenerative diseases [67]; voltage-gated and ligand-gated potassium channels (*kcnc1a* and *kcnma1a*), which are vital for proper voltage conductance in neuronal cells [96,97]; and GRIN family members (*grin1a* and *grin2ca*), which are required for neurotransmission [68]. Additionally, the term “synaptic signaling” was shared by all conditions except for the 0.9 μg/L dose, with the 0.03 μg/L conditions again exhibiting the most pronounced enrichment, and highlighted genes such as *cdh2* and *slc17a6b*, *slc6a5*, and *slc6a2* (Appendix A), which are genes for glutamate receptors [98]. These observations suggested lasting deficits in the molecular machinery necessary for neurotransmitter release and receptors. Similarly, studies in mouse cortical neurons have demonstrated that citalopram exposure leads to a dose-dependent inhibition of neuronal firing frequency, an effect mediated by rectifying potassium channels [99].

The term “glutamate receptor” was identified in all conditions, except 250 μg/L citalopram (Appendix A). Although GRIN family genes were shared among the sets, the 0.03 μg/L dose exhibited significantly stronger negative enrichment for genes such as *gria1a* and *gria4a*, indicating that excitatory neurotransmission may be broadly altered well after exposure. Complementary DNA methylation analyses in citalopram treated human embryonic stem cell-derived telencephalic neurons have revealed non-linear, time-dependent methylation changes in key synaptic and neuronal genes, including *cdh2*, potassium channels (*kcnd3*), glutamate receptors (*gria1* and *gria4*), and various SLC genes which demonstrates a dynamic regulation of these components during neuronal differentiation [58].

GSEA identified downregulated gene groups linked to axonal development and function in all conditions except for 0.03 μg/L, with the 50 μg/L dose showing the greatest enrichment (Appendix A). Shared genes with consistently robust negative enrichment across conditions include *cntn3a.1* (contactin 3a), a mediator of cell–cell adhesion, and TH (tyrosine hydroxylase), which catalyzes L-DOPA production and is critical for dopamine biosynthesis and spinal cord motor neuron differentiation (Appendix A) [100]. Another shared term, present in all conditions except for 0.03 μg/L, is “somatodendritic compartment” (Appendix A), which comprises genes associated with the neuronal cell body and dendrites. Within this category, microtubule associated proteins *map1aa* and *map1b* are critical for cytoskeletal organization (Appendix A). Collectively, these findings indicated the discontinued citalopram exposure disrupted multiple organizational levels of neuronal function, affecting the formation and stability of synaptic membranes as well as the regulation of neurotransmitter signaling and overall neuronal integrity. TF target analysis indicated that the consistent downregulation of synaptic and neuronal gene sets across citalopram concentrations was closely linked to disruptions in key transcription factors including NRSF, FOXM1, and AHRARNT [86,101,102]. These TF changes were driven by downregulated genes such the glutamate receptor subunit, *grin1a*, the synaptic adhesion molecule, *cdh2*, and potassium and calcium channel genes (Appendix A). These target pathways may provide a plausible mechanism by which discontinued citalopram exposure persisted in impairing synaptic organization and neuronal function.

All gene sets discussed above held robust negative enrichment scores of at least 1.5, implying that even short-term alterations to serotonergic signaling during critical early developmental windows such as the first 24 hpf of zebrafish development had enduring effects on neuronal and synaptic function. These synaptic and neuronal impairments could underlie deficits in circuit formation and plasticity, potentially offering mechanistic insight into how developmental SSRI exposure leads to later neurobehavioral changes.

### 3.5. Metabolic and Stress Response

SSRIs, including citalopram, have been shown to inhibit components of the mitochondrial electron transport chain, particularly complexes I and IV [103]. When oxidative phosphorylation is impaired in localized regions of the brain (ex: frontal cortex and hippocampus), decreased ATP production occurs, which has been implicated as a causative factor in bipolar disorders [103]. In our study, altered serotonergic signaling following citalopram exposure was associated with dose-dependent enrichment of metabolic pathways. At the lowest dose (0.03 μg/L), these included upregulation of oxidative phosphorylation and the TCA cycle driven by genes such as *atp8*, *cox2*, *atp6v0ca*, *ndufa1* as well as glycine/serine/threonine metabolism driven by genes such as *grhprd*, *gatm*, *shmt2* (Appendix A). These pathways have been linked to neuronal growth, mitochondrial function, and synaptic activity [104,105,106]. Mitochondrial dysfunction has also been associated with depressive disorders in various citalopram exposure studies [58,107,108].

Interestingly, various studies have proposed a pro-neurogenesis/synaptogenesis function for SSRIs (including citalopram) specifically in disease models [109,110]. In Alzheimer’s disease neurons, treatment with citalopram improved synapse and neuron formation, while also enhancing cell survival and mitochondrial respiration [110]. In our study, lower citalopram concentrations were associated with upregulation of metabolic pathways that, in other contexts, have been linked to neuronal growth and function [109]. At higher doses, these same pathways were accompanied by strong enrichment for stress, and apoptosis-related processes, suggesting a shift toward cellular stress responses (Appendix A). The varied responses found in other studies of citalopram exposure are reflected in the non-linear responses uncovered by our RNAseq analysis.

Extremely low doses of citalopram (0.03 μg/L) elicited pronounced expression changes in genes associated with metabolic functions, alterations that are partially shared with higher doses, yet become lost in favor of metabolic shifts indicative of stress responses and apoptotic mechanisms. One common process shared by 0.03 μg/L and 50 μg/L citalopram exposures is the regulation of oxidoreductase function, with both conditions showing significant regulatory enrichment in this pathway. Although the most upregulated genes differ between the two doses, the overarching process is mutual. At 0.03 μg/L, the GO term “oxidoreductase complex” was positively enriched (Appendix A). This enrichment was primarily driven by several genes associated with mitochondrial respiration and metabolic processes which include: alpha-ketoglutarate dehydrogenase subunit 4 (*kgd4*), a critical enzyme in the Krebs cycle; NADH ubiquinone oxidoreductase subunit A1 (*ndufa1*), a core component of complex I in the electron transport chain; and branched-chain keta acid dehydrogenase E1 alpha subunit (*bckdha*), which participates in the catabolism of branched chain amino acids (Appendix A) [111,112,113,114]. Additional genes that contributed to the “oxidoreductase complex” term included glycerol-3-phosphate dehydrogenase 1c (*gpd1c*), involved in respiration, glycolysis, and lipid metabolism, and NADH ubiquinone oxidoreductase subunit B9 (*ndufb9*), another subunit of complex I (Appendix A) [112,114]. In contrast, at 50 μg/L, the enriched terms include “oxidoreductase activity, acting on peroxide as acceptor,” driven by genes such as prostaglandin endoperoxide synthase 2a (*ptgs2a*; catalyzes the synthesis of prostaglandin), glutathione peroxidase 4a (*gpx4a*; involved in the oxidative stress response via glutathione peroxidase activity), microsomal glutathione S-transferase 3b (*mgst3b*; important for glutathione and leukotriene metabolism), and peroxiredoxin 5 (*prdx5*; critical for regenerative processes) (Appendix A) [92,115,116]. Additionally, the term “oxidoreductase activity, acting on CH-OH as donors” was enriched (Appendix A), driven by phosphoglycerate dehydrogenase (*phgdh*; key for L-serine biosynthesis) and sorbitol dehydrogenase (*sord*; involved in sorbitol catabolic processes) (Appendix A) [117,118]. These shared metabolic processes indicate that citalopram-induced alterations in oxidoreductase function may disrupt cellular redox balance and energy metabolism, thereby potentially compromising cell viability and function during development. The dysregulation of these metabolism enzymes could impair mitochondrial efficiency and energy production.

Gene set enrichment analysis (GSEA) further revealed that the 0.03 μg/L exposure is associated with unique, dose-specific, metabolic processes. At 0.03 μg/L, there was enrichment for the “oxidative phosphorylation” pathway (Appendix A). This enrichment was driven by the differential expression of several key genes, including ATPase phospholipid transporting 8A1 (*atp8*), which is critical for ATP synthesis; cytochrome c oxidase assembly factor COX2 (*cox2*), a component of complex IV of the electron transport chain; ATPase H+ transporting v0 subunit ca (*atp6v0ca*), essential for V-ATPase-dependent organelle acidification; NADH ubiquinone oxidoreductase subunit A1 (*ndufa1*), part of complex I of the electron transport chain; and ATP synthase peripheral stalk-membrane subunit b (*atp5pb*), a subunit of ATP synthase (Appendix A) [78,119,120,121,122,123]. Enrichment was also observed in the “TCA cycle” pathway (Appendix A), driven by genes such as pyruvate carboxylase a (*pcxa)*, which catalyzes the formation of oxaloacetate necessary for the Krebs cycle; isocitrate dehydrogenase (NAD+) 3 non-catalytic subunit (*idh3g*), an important subunit in the enzyme complex responsible for producing 2-oxoglutarate; and malate dehydrogenase 1Aa (*mdh1aa*), which plays a key role in carbon metabolism, lipid metabolism, and redox balance (Appendix A). Additionally, there was enrichment in the “glycine, serine, and threonine metabolism” pathway, driven by glyoxylate reductase/hydroxypyruvate reductase b (*grhprb*), glycine amidinotransferase (*gatm*), and serine hydroxymethyltransferase 2 (*shmt2*) (Appendix A).

In contrast to the 0.03 μg/L condition, at 0.9 μg/L, unique enrichments were observed for processes such as “tetrapyrrole binding” and “arachidonic acid metabolism” (Appendix A), both of which prominently feature prostaglandin-endoperoxide synthase 2a (*ptgs2a*; catalyzes prostaglandin synthesis) as a strongly enriched gene (Appendix A) [115]. These distinct metabolic signatures suggest that dose-dependent effects of citalopram may differentially modulate pathways involved in cellular energy production, amino acid metabolism, and lipid metabolism. Collectively, the shared and unique metabolic alterations underscore a mechanistic link between disrupted metabolic processes and impaired neuronal development, as alterations in these critical pathways may compromise the formation and maintenance of synaptic structures. This may ultimately lead to impaired neuronal connectivity and synaptic function.

At higher doses (50 μg/L and 250 μg/L), there is a clear shift toward stress response pathways, accompanied by an increase in apoptosis-associated gene expression. At 50 μg/L, the “apoptotic signaling pathway” term is significantly enriched (Appendix A), driven by genes such as tumor protein p53 (*tp53*; key regulator of cell cycle arrest, DNA repair, and cell death pathways), tumor protein p63 (*tp63*; modulates cell proliferation, differentiation, and senescence), death-associated protein (*dap*; interferon dependent inducer of cell death pathways), MCL1 apoptosis regulator (*mcl1b*; pro-survival Bcl2 type), and cathepsin Lb (*ctslb*; cysteine protease involved in programmed cell death) (Appendix A) [124,125,126,127,128]. In contrast, at 250 μg/L, the enriched term “apoptosis” is driven by a distinct set of genes (Appendix A), including mitogen-activated protein kinase 5 (*map3k5*; important in stress response and cell death), baculoviral IAP repeat containing 2 (*birc2*; inhibitor of apoptosis), actin beta 1 (*actb1*; cytoskeletal component important for cell growth and migration), BCL2 like 1 (*bcl2l1*; anti-apoptotic protein), lamin A (*lmna*; structural protein of the nuclear lamina), and Jun proto-oncogene (*jun*; linked to tumor-suppression and inflammation response) (Appendix A) [129,130,131,132,133,134].

Further adding to the shift into a stress response state, both conditions exhibit co-enrichment of metabolic and toxic stimulus response terms. At 50 μg/L, for instance, “glutathione metabolism” is enriched (Appendix A)—driven by glutathione peroxidase 4a (*gpx4a*; involved in the oxidative stress response via glutathione peroxidase activity), microsomal glutathione S-transferase 3b (*mgst3b*; important for glutathione and leukotriene metabolism), and ChaC cation transport regulator 1 (*chac1*; endoplasmic reticulum stress inducible involved in glutathione metabolism)—along with “response to toxic stimulus” involving prostaglandin-endoperoxide synthase 2a (*ptgs2a*; catalyzes the synthesis of prostaglandin) and *gpx4a* and “mitochondrial envelope” with contributions from solute carrier family 25 member 25a (*slc25a25a*; mitochondrial transporter involved in ciliary function) and serine hydroxymethyltransferase 2 (*shmt2*; catalyzes synthesis of glycine and critical for one carbon metabolism) (Appendix A) [92,115,128,135,136]. At 250 μg/L, the term “cellular response to external stimulus” is enriched (Appendix A), driven by KN motif and ankyrin repeat domains 2 (*kank2*; cytoskeletal/adhesion binding protein), GABA receptor associated protein b (*gabarapb*; associated with interferon related host defense), and again *dap* (Appendix A) [126,137,138].

High-dose citalopram exposure has been similarly implicated in multiple cell models. For instance, in rat hepatocytes, high concentrations of citalopram produced increased cell death, elevated ROS formation, mitochondrial dysfunction, lysosomal membrane leakiness, lipid peroxidation, and glutathione depletion, all of which are consistent with our gene expression analysis [139]. At higher doses, some studies have indicated that there is a variable effect of cytotoxicity of citalopram on different cell types [140]. For example, Schwann cells are relatively unaffected whereas neuroblastoma cells are impaired [140]. Collectively, these enriched terms and phenotypic observations indicate that at higher doses of citalopram, cells shift toward a stress response state characterized by activation of apoptotic pathways and metabolic adaptations. Such changes may compromise energy homeostasis and potentially affect synaptic integrity, which could manifest as impaired neurodevelopment. This suggests that the activation of these pathways could underlie the neurobehavioral deficits following developmental SSRI exposure by disrupting the balance between survival and cell death during critical periods of development during which time it is possible that citalopram influences particularly susceptible cell types. The transcription factor target analysis aligned with the observed dose-dependent metabolic disruptions. At low doses, enrichment of stress-responsive factor gene sets such as SQSTM1 (Table 1) suggested compensatory responses such as increased autophagic flux [141], while high-dose exposures were marked by negative enrichment of protective regulators such as OTC1 and SMCHD1, consistent with impaired metabolic protection [81,87]. These findings reinforced the broader transcriptomic pattern of mild stress adaptation at low doses and pronounced oxidative and apoptotic signaling at higher concentrations of citalopram.

### 3.6. Developmental and Morphogenic Processes

Multiple doses of citalopram tested in this study yielded the term “structural constituent of eye lens” as an enriched gene set, except at the highest dose (250 μg/L). The lowest dose (0.03 μg/L) yielded a clear upregulatory enrichment of multiple crystallin gamma 2d genes (*crygm2d1*, *5*, *7*, *9*, *10*, *12*, *13*, *18*, and *19*; all structural components of eye lens fiber cells) [142], all of which showed enrichment scores above 2 (Appendix A). While similar patterns were observed at 0.9 and 50 μg/L, none of the crystallin genes reached enrichment scores above 1.5 and 1, respectively.

Ectodermal placodes are discrete cranial ectodermal thickenings in developing vertebrates that give rise to various ectoderm-derived structures. This includes placodes for olfactory, lens, adenohypophyseal, otic, ophthalmic, and other sensory systems [143]. Of particular relevance are the lens and otic placodes, which differentiate into the lens of the eye and the inner ear, respectively [144,145]. The 0.03 μg/L dose exhibited “ectodermal placode development” as an enriched gene set (Appendix A), largely driven by the upregulation of paired Box 2a (*pax2a*), which is important for nervous system development, including eye and ear development, and the starmarker gene (*stm*), which is crucial for inner ear and sensory system development (Appendix A) [146,147,148,149]. During its development, the lens placode modulates ECM formation, particularly collagens, partly via the regulation of metalloproteases [150]. At 0.03 and 50 μg/L citalopram, we observed upregulation of the “collagen trimer” gene set, driven primarily by collagens I, IV, and VII, alongside downregulation of the “cell–cell adhesion” gene set, attributed to changes in uncharacterized non-coding RNA LOC103911741, contactin 3a (*cntn3a.1*), and protocadherin genes (*pcdh1* and *pcdh2* subtypes) (Appendix A) [151,152,153]. Disruption of ECM composition and adhesion molecules can compromise lens fiber cell structure, potentially inducing cellular stress responses [154]. Notably, serotonin signaling at high concentrations has been shown to induce alpha crystallin aggregation and alter alpha crystallin hydrophobicity and oligomer size [155], changes that can destabilize lens fiber cells and make them more susceptible to apoptosis. Indeed, exposure to selective serotonin reuptake inhibitors (SSRIs) has been linked to ocular complications, possibly through apoptotic pathways in lens fiber cells [156,157]. Thus, the ECM and adhesion changes observed here may represent upstream events that predispose lens fiber cells to the types of apoptotic effects reported in other SSRI exposure models. Further, transcription factor disruptions at 0.03 μg/L—including altered activity of ZMYND11, HOXC13, and ZNF586—may further dysregulate lens ECM and crystallin gene expression through effects on targets such as *col1a1*, *ddr1*, *junba*, *stat5b*, *kdm4b*, and *ank2a/b* which may compound the developmental sensitivity of the lens placode to citalopram exposure.

Our data demonstrate the upregulation of crystallin gamma 2d genes, the enrichment of ectodermal placode-related gene sets, and the modulation of collagen and adhesion pathways. These results imply that serotonin signaling may modify key processes involved in lens placode formation. It is possible that serotonin signaling modulates the lens placode itself, which in turn influences the expression and organization of critical collagens and adhesion molecules necessary for proper lens morphogenesis. This aligns with evidence linking SSRI exposure, including low doses of citalopram, to ocular defects and supports the notion that serotonin signaling is integral to normal lens and eye development in vertebrates [158,159,160].

## 4. Materials and Methods

### 4.1. Zebrafish Maintenance and Husbandry

Adult WT *Danio rerio* (zebrafish) were obtained from Carolina Biologicals (Burlington, NC, USA). They were maintained under standard conditions in fish water at 28 °C with a 14 h light/10 h dark cycle. Dishes filled with marbles were placed into breeding tanks the evening before embryo collection, and the fish were allowed to remain undisturbed overnight. All procedures were conducted under protocols approved by the Institutional Animal Care and Use Committee at Montana State University.

### 4.2. Zebrafish Embryo Treatment and RNA Extraction

Zebrafish embryos were collected shortly after fertilization and allocated to either the control group (0.3× Danieau) or one of four treatment groups exposed to citalopram hydrobromide (M.W. 405.3 g/mol; Millipore Sigma C7861, St. Louis, MO, USA) dissolved in 0.3× Danieau. The tested concentrations—0.03, 0.9, 50, and 250 μg/L—correspond to measured levels in environmental surface water [161], wastewater treatment plant effluent [161], therapeutic human serum [162], and super-therapeutic doses [162], respectively. Prior to treatment, embryos were cleaned thoroughly, and only healthy, viable embryos were included in the experimental setup. Exposure commenced at approximately 2 h post-fertilization (hpf) and continued until 24 hpf, after which embryos were rinsed thoroughly with 0.3× Danieau to remove residual citalopram and maintained at 28.5 °C in clean 0.3× Danieau until harvest at 48 hpf (Figure 4A). This exposure paradigm was designed to identify persistent molecular effects following transient early developmental perturbation. Each condition included five biological replicates consisting of 50 pooled embryos per replicate, which were processed for RNA sequencing and subsequent bioinformatic analysis (Figure 4B).

The RNA extraction process was adapted from the protocol of Peterson and Freeman and as previously described (Figure 1A,B) [55,163]. In brief, 50 embryos (with excess liquid removed) were placed in nuclease-free 1.5 mL microcentrifuge tubes and lysed in TRIzol reagent (Thermo Fisher Scientific, Cat. No. 15596026, Carlsbad, CA, USA). The embyros were homogenized with 40–50 strokes using a P1000 pipette tip, followed by vigorous vortexing until complete lysis was achieved. Chloroform was added to the homogenate to separate the phases, and the RNA-enriched aqueous phase was collected. RNA was precipitated using isopropanol, and the resulting pellet was washed three times with ethanol before being resuspended in nuclease-free water. A DNase I treatment (Thermo Fisher Cat. No. EN0521, Carlsbad, CA, USA) was performed per the manufacturer’s guidelines, and the samples were subsequently purified using the Monarch RNA Cleanup kit (New England Biolabs, Cat. No. T2030L, Ipswich, MA, USA).

RNA quality was assessed with a Nanodrop spectrophotometer and an Agilent Bioanalyzer 2100 (using the Agilent RNA 6000 Nano Kit, Cat. No. 5067-1511, Santa Clara, CA, USA). Samples displaying RNA Integrity Numbers (RINs) of 7 or higher and A260/A280 ratios between 1.8 and 2.0 were used for further analysis (*n* = 5). These high-quality RNA samples were sent to the University of Montana Genomics Core (UMGC) for cDNA library preparation using the Zymo-Seq RiboFree Total RNA Library Kit (Cat. R30003, Irving, TX, USA). Sequencing was performed on the Illumina NovaSeq X Plus platform (Illumina, San Diego, CA, USA) using a paired-end approach with 150 bp reads, targeting roughly 30 million reads per sample and yielding approximately 10× coverage.

### 4.3. Software and Computational Pipeline

Initially, the raw paired FASTQ files were subjected to quality assessment using FASTQC (version 0.11.9-Java0-11) [164]. This was followed by error correction and adapter trimming using rCorrector (version 1.0.7) [165] and Trimmomatic (version 0.39) [166], respectively. In the Trimmomatic step, adapter sequences were removed using parameters 2:30:10:2, with additional trimming of low-quality leading and trailing bases (threshold of 3) and a minimum read length requirement of 36 bp. After trimming, average reads retained for control, 0.03, 0.9, 50, and 250 μg/L was 98.60%, 98.6%, 98.67%, 98.63%, and 98.59%, respectively, with a minimum Phred score of 30 enforced to ensure data quality. Cleaned reads were then mapped and quantified using Salmon (version 1.10.1) [167] against a reference library that included both coding and non-coding zebrafish RNA sequences, supplemented with decoy genomic sequences from the GRCz11 Ensembl assembly [168]. The counts from each sample were compiled into a master matrix and analyzed for differential expression using DESeq2 as part of the Trinity RNA-Seq framework (version 2.15.2) [169].

For PCA and PLS-DA, standardized transcript per million (TPM) data derived from Salmon were imported into R (version 4.3.2) [170]. Genes with zero variance were removed to prevent computational artifacts. For PCA, principal components were calculated via prcomp function within basic R functionality. This was visualized via plotting pairwise PCA plots via ggplot2 [171] with 95% confidence interval ellipses. For PLS-DA, mixOmics package was used to model sample groupings based on condition type, and the resulting analysis was visualized via mixOmics and ggplot2 with ellipses corresponding to 95% confidence intervals [172].

### 4.4. Overrepresentation Analysis (ORA) and STRING Analysis

Differentially expressed transcripts (DETs) were identified by filtering expression matrices to retain transcripts with an absolute log_2_-fold change > 0.6 (approximately a 1.5-fold change) and an adjusted *p*-value ≤ 0.05. Transcripts meeting both criteria were classified as DETs. Significant DETs were visualized via volcano plots in R via ggplot2 [171]. Genes were categorized as upregulated (red) or downregulated (blue) based on the directionality of their log_2_FC value. Dashed lines were added to indicate the significance thresholds.

For pathway and gene ontology (GO) term enrichment analysis, the DETs were first converted to corresponding gene identifiers using the g:Convert tool in g:Profiler (version e111_eg58_f463989d) now referred to as DEGs [173]. The resulting gene list was then submitted to WebGestalt (version 2024) to perform overrepresentation analysis (ORA) for KEGG pathways, transcription factor targets, and GO categories—Biological Process, Cellular Component, and Molecular Function—using the noRedundant option to minimize redundant terms. The analysis was conducted against the affy zebgene 1.1 st v1 reference set, employing a weighted set cover method for redundancy removal and adjusting for multiple comparisons via the Benjamini–Hochberg method [174]. ORA is employed to determine highly overrepresented functions based on the already filtered gene set, yet may overlook coordinated changes involving genes with lower expression, for which GSEA was used.

In parallel, functional network analysis for transcription factor targets was executed using the same significance thresholds (*p*-value ≤ 0.05 and FDR ≤ 0.05) in WebGestalt. To further corroborate pathway interactions, the DEGs were also analyzed using STRING (version 12.0) [175]. DEG lists were organized into upregulated, downregulated, and combined sets, with each set submitted independently for STRING analysis [175]. STRING performed its own enrichment (including KEGG, GO, and local cluster analysis) using a minimum required interaction score of 0.400 (medium confidence) and an FDR stringency of 5%; terms meeting these criteria were considered significantly enriched. While STRING offers insight into the physical and functional interactions within the submitted DEG list, it remains dependent on the initial DEG selection and fails to capture the full spectrum of subtle expression changes.

### 4.5. Gene Set Enrichment Analysis (GSEA)

In contrast to the threshold-dependent approaches of ORA and STRING analysis, gene set enrichment analysis (GSEA) was employed to capture broader, coordinated biological trends that may have been overlooked by ORA and STRING. Transcript-level output from DESeq2 was converted to corresponding gene Entrez IDs using the g:Convert tool in g:Profiler [173], and in cases where multiple transcripts mapped to a single gene, only the transcript with the highest *p*-value was retained with duplicates removed. A ranked list of genes and their corresponding local scores—calculated as:Local Score = SIGN(log_2_FC) × log_10_ of the raw *p*-value
was then submitted to WebGestalt’s GSEA function [174], configured with 5000 permutations, an enrichment statistic of 1, and a range of 5 to 2000 analytes per category [51]. Redundancy among enriched gene sets was minimized using the Weighted Set Cover function in WebGestalt, and enrichment analyses were conducted for KEGG pathways as well as Gene Ontology categories including Biological Process noRedundant, Cellular Component noRedundant, Molecular Function noRedundant, and Transcription Factor Targets. GSEA is designed to detect subtle biologically meaningful shifts in expression across entire gene sets in bulk data, thereby complementing the discrete pathways from ORA and STRING analysis.

### 4.6. Single-Cell Deconvolution of Bulk RNAseq

To determine whether citalopram treatments altered the proportions of cell types in whole embryos, we performed single-cell deconvolution of our bulk RNAseq data. This approach uses known single-cell transcription reference profiles to estimate cell type abundances from bulk gene expression patterns, enabling comparison of relative cell type proportions across samples.

Bulk RNA-seq single-cell deconvolution was performed in R using MuSiC (ver.2) [176] to estimate cell type proportions in zebrafish bulk RNA data. The single cell reference was derived from Atlas 1.0 of zebrafish development [177], and was restricted to cell types present at 48 h post-fertilization (hpf). Only clusters (cell types) with at least two cells in each type category within the Atlas 1.0 dataset were retained. An ExpressionSet was constructed from these data and converted into a SingleCellExperiment object for use in MuSiC. Bulk RNA-seq data, normalized as transcript per million (TPM), were pre-processed by aggregating duplicate gene entries (transcripts mapping to a single gene causing duplicate values) via averaging and by filtering out genes with a mean TPM below 1. Low-expression genes (aggregate count values of 1 or less across replicates) were excluded from the bulk-RNA TPM dataset. R (v4.3.2) packages used for the single cell deconvolution included: MuSiC [178], Biobase [178], and SingleCellExperiment [179]. Cell Type proportions output by MuSiC for every sample (*n* = 5 per citalopram condition) were normalized with arcsin square root normalization, after which they were compared via two-tailed Welch’s *t*-test. *p*-values were FDR adjusted via Benjamini–Hochberg procedure to produce adjusted *p*-values [174].

## 5. Conclusions

Zebrafish embryos were exposed to a range of citalopram concentrations from 2 to 24 hpf, followed by a wash and subsequent move to citalopram-free conditions until 48 hpf. Following exposure, whole-embryo RNA sequencing revealed persistent, dose-specific shifts in gene expression. Our data show that even short-term perturbation of serotonergic signaling during the first 24 h of zebrafish embryo development elicited significant changes in gene expression, with the lowest dose producing the most differential expression and intermediate doses producing the least. Consistent across all concentrations was the downregulation of genes related to synaptic architecture and neurotransmission. Concurrently, two distinct metabolic disruptions emerged: at low concentrations, we observed upregulation of metabolic pathways (glycine/serine/threonine metabolism), whereas higher doses drove stress-related processes (glutathione metabolism, ferroptosis) and apoptotic signaling. Interestingly, low-dose citalopram exposure significantly influenced eye-lens, extracellular matrix, and cell adhesion gene sets, suggesting broad developmental repercussions for lens morphogenesis and tissue organization. Single-cell deconvolution showed that these transcriptional changes occurred independently of major shifts in cell-type composition, indicating persistent regulatory changes within stable developmental cell populations. Together, these findings demonstrate that transient serotonergic interference produces lasting, dose-dependent impacts on embryonic development. Further, our findings demonstrate the zebrafish model’s translational relevance for studying human disorders linked to disrupted neural circuit formation, metabolism, and tissue organization.

## Figures and Tables

**Figure 1 ijms-26-11288-f001:**
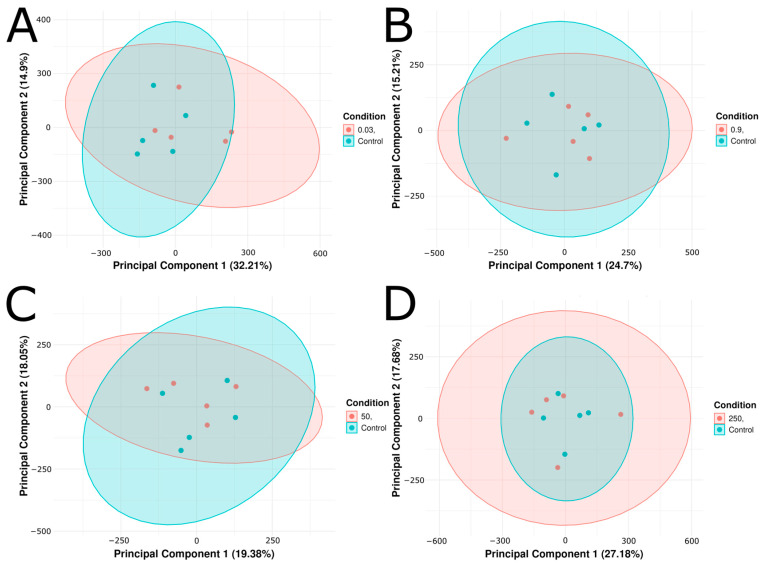
Principal component analysis (PCA) plots of (**A**) Control vs. 0.03 μg/L citalopram, (**B**) Control vs. 0.9 μg/L, (**C**) Control vs. 50 μg/L, and (**D**) Control vs. 250 μg/L citalopram. Each plot shows the first two principal components derived from transcript per-million (TPM) normalized gene expression values. Five biological replicates (*n* = 5) per condition are shown as individual points, and ellipses correspond to 95% confidence intervals. The percentage of variance explained by each principal component is displayed on the x and y axis.

**Figure 2 ijms-26-11288-f002:**
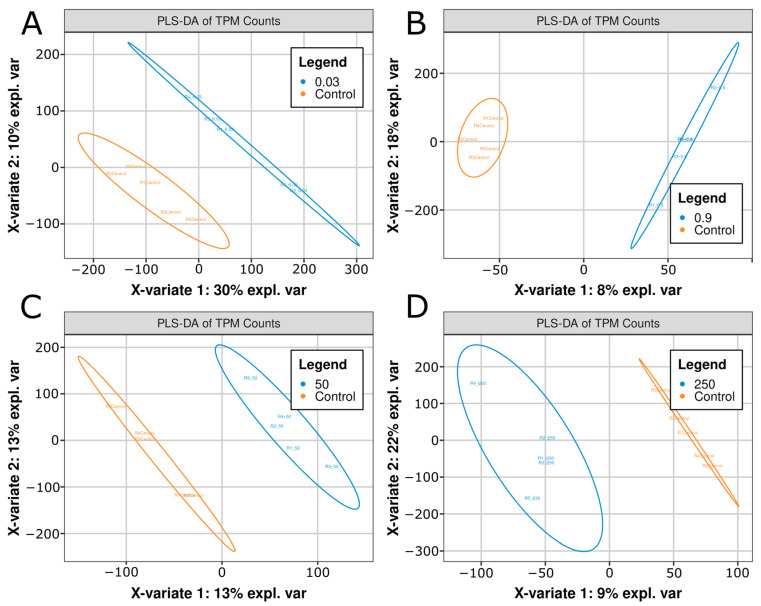
Partial least squares discriminant analysis (PLS-DA) plots of (**A**) Control vs. 0.03 μg/L citalopram, (**B**) Control vs. 0.9 μg/L, (**C**) Control vs. 50 μg/L, and (**D**) Control vs. 250 μg/L citalopram. Each plot shows the separable x and y variates derived from transcript per-million (TPM) normalized gene expression values. Five biological replicates (*n* = 5) per condition are shown as individual points, and ellipses correspond to 95% confidence intervals. The percentage of variance explained by each principal component is displayed on the x and y axis.

**Figure 3 ijms-26-11288-f003:**
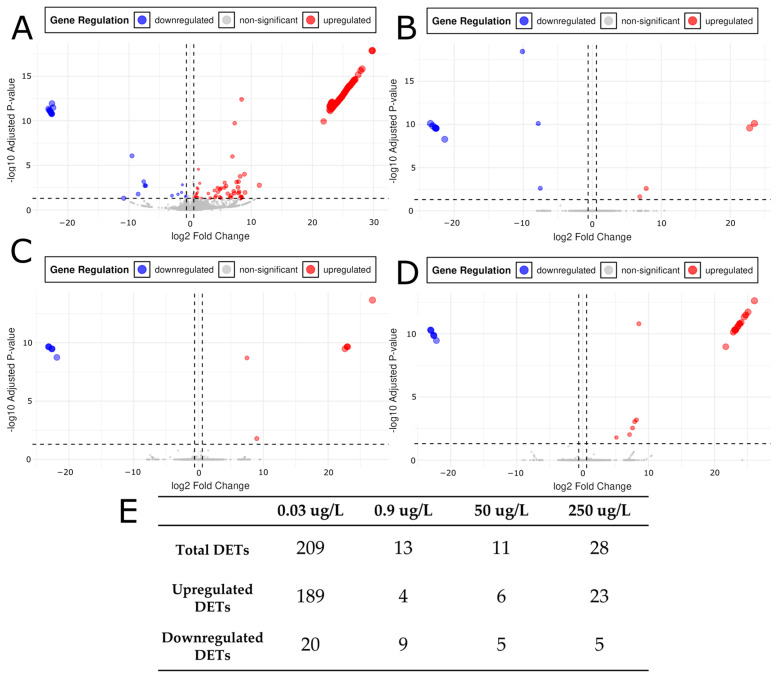
Volcano plots for (**A**) Control vs. 0.03 μg/L citalopram, (**B**) Control vs. 0.9 μg/L, (**C**) Control vs. 50 μg/L, and (**D**) Control vs. 250 μg/L citalopram. X axis threshold dotted lines occur at ±0.6 log_2_FC. This requires an expression change of ±1.5 compared to control. The Y axis threshold dotted line occurs at p_adj_ = 0.05. Red colorations are differentially expressed transcripts (DETs) with a positive log_2_FC and significant p_adj_ (≤0.05). Blue coloration is indicative of differentially express transcripts with a negative log_2_FC and significant p_adj_ (≤0.05). (**E**) DETs relative to controls reported for each concentration of citalopram tested (0.03, 0.9, 50, and 250 μg/L) for total, upregulated, and downregulated subcategories.

**Figure 4 ijms-26-11288-f004:**
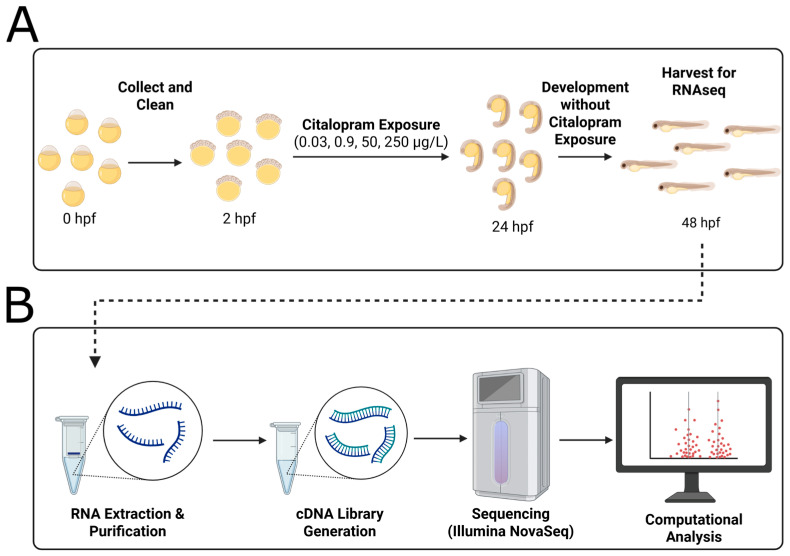
Citalopram hydrobromide exposure protocol and RNA sequencing protocol. (**A**) Exposure protocol. Zebrafish embryos were collected and cleaned immediately after fertilization. Exposure to citalopram (0.03, 0.9, 50, or 250 μg/L, corresponding to wastewater effluent, surface water, therapeutic blood serum, and super-therapeutic concentrations, respectively [161,162]) began at 2 h post-fertilization (hpf) and continued until 24 hpf. After exposure time, embryos were washed and allowed to develop in clean 0.03× Danieau solution until 48 hpf at which point pooled embryos (50 per replicate; *n* = 5 per condition) were collected for (**B**) RNA extraction and sequencing. cDNA libraries were prepared and sequenced on Illumina NovaSeq X platform, and subsequent bioinformatic analyses were conducted internally.

**Table 1 ijms-26-11288-t001:** Significant transcription factors identified from transcription factor target pathways via gene set enrichment analysis (GSEA) input for each citalopram concentration (0.03, 0.9, 50, and 250 μg/L). For each transcription factor, the total number of genes in each set is reported in the size column, and the number of enriched input terms driving transcription factor enrichment is reported in the LEN column. LEN is the leading-edge number and indicative of how many genes in the input set are altered. Magnitude and direction of change in a gene set is reported as the normalized enrichment score in the NES column. Finally, the *p*-value and false discovery rate (FDR) are reported in their respective columns.

	Transcription Factor	Size	LEN	NES	*p* Value	FDR
0.03 μg/L	ZNF418	179	92	−2.2985	<2.2 × 10^−16^	<2.2 × 10^−16^
	ZNF586	204	96	−2.3364	<2.2 × 10^−16^	<2.2 × 10^−16^
	SMCHD1	183	99	−2.4001	<2.2 × 10^−16^	<2.2 × 10^−16^
	MYF6	143	91	−2.5288	<2.2 × 10^−16^	<2.2 × 10^−16^
	BAHD1	168	94	−2.563	<2.2 × 10^−16^	<2.2 × 10^−16^
	EGFR	90	47	1.936	<2.2 × 10^−16^	0.001727
	SQSTM1	129	76	1.9149	<2.2 × 10^−16^	0.001832
	ZMYND11	72	30	1.9033	<2.2 × 10^−16^	0.001902
	HOXC13	236	99	1.8735	<2.2 × 10^−16^	0.002177
	RUVBL2	96	57	1.7621	<2.2 × 10^−16^	0.00944
0.9 μg/L	STN1	20	9	1.8668	0.001223	0.017524
50 μg/L	NRSF_01	243	110	−2.5719	<2.2 × 10^−16^	<2.2 × 10^−16^
	HNF_16	619	193	−2.1751	<2.2 × 10^−16^	4.04 × 10^−5^
	CDP_02	475	200	−2.198	<2.2 × 10^−16^	4.49 × 10^−5^
	OCT1_02	563	215	−2.2005	<2.2 × 10^−16^	5.05 × 10^−5^
	AHRARNT_01	512	204	−2.2006	<2.2 × 10^−16^	5.77 × 10^−5^
	OCT1_06	712	240	−2.2073	<2.2 × 10^−16^	6.73 × 10^−5^
	OCT1_04	562	229	−2.2123	<2.2 × 10^−16^	8.08 × 10^−5^
	FOXM1_01	778	283	−2.2299	<2.2 × 10^−16^	0.000101
	MED25	304	67	1.6288	<2.2 × 10^−16^	0.03825
	PSMB5	622	170	1.6518	<2.2 × 10^−16^	0.045747
250 μg/L	MYF6	146	88	−2.2294	<2.2 × 10^−16^	<2.2 × 10^−16^
	MSX2	148	89	−2.2541	<2.2 × 10^−16^	<2.2 × 10^−16^
	HSF4	98	66	2.0582	<2.2 × 10^−16^	0.000404
	SMCHD1	188	78	−2.1193	<2.2 × 10^−16^	0.00067
	ZNF510	205	93	−2.0818	<2.2 × 10^−16^	0.001392
	ZNF202	333	127	−2.0446	<2.2 × 10^−16^	0.001439
	SOX11	204	81	−2.0405	<2.2 × 10^−16^	0.001514
	ZA_UNIPROT_Q9UM89	233	88	1.6495	0.000565	0.045507

## Data Availability

Raw fq.gz files for all sequencing data, count tables output from Salmon processes, and GSEA tables can be found at NCBI Gene Expression Omnibus under the accession number GSE308723.

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
