# Peer review of "Persistent and Dose-Dependent Neural and Metabolic Gene Expression Changes Induced by Transient Citalopram Exposure in Zebrafish Embryos"

_ijms, 2025, doi:10.3390/ijms262311288_

Round 1
Reviewer 1 Report
Comments and Suggestions for Authors
Overall, this is a great manuscript that adds significant and valuable information to the field. I appreciate the amount of work that has gone into this, both scientifically and in writing. The introduction is well written and addresses all points of interest. It might be helpful to include a diagram of the interactions and metabolic products (this is absolutely a bonus request, and the manuscript works well without, but if you feel that you have the time for it, that might help some readers understand the processes, the relevance, and guide the interpretation of the results). The results are well written, but a lot of the sections begin with what reads like a summary of the method. I suggest removing these sections entirely or re-using them in the methods. The results section should focus only on the data that was generated, and the visualisation thereof. The discussion is extensive (sometimes too much so) and would benefit from cutting down to the main messages. I appreciate that there is a lot to discuss, but I believe that the strongest messages are lost in the plethora of information. The methods are not very clear in parts. I suggest providing more detail on experimental set up and the animal housing/husbandry and how the embryos were euthanised prior to analysis.
All comments I have made are for further improvements of an already great manuscript.
Specific comments:
Line 42: Please include the environmentally relevant concentration (range) here.
Line 48: No need to re-abbreviate SSRIs, as that was already done in line 36.
Line 57: what form of tryptophan are you referring to? Please either write ‘L-tryptophan’ or use the abbreviation for TPH1.
Line 58: TPH2 is not explained (I appreciate that it follows the logic of TPH1, but I would still suggest writing it out the first time).
Line 85ff: Consider referring to a review paper on ZFE as an alternative test system, as this paragraph is rather short in comparison to the others.
Lines 92-100: As all results are summarised from publication 45, consider re-phrasing to clarify this and use the citation only once.
Line 113ff: Would it be possible to include something in the previous paragraph that identifies the exposure regime as a limiting factor to current knowledge. This is the first time that you mention your planned exposure duration and recovery period, and it is not clear (yet) why. Note: I am not questioning the set-up, I believe that this is a great approach. This comment is just meant to highlight that even more.
Lines 126ff: This reads like the start of your methods. I believe this (and figure 1) should be moved into the methods section instead (up until “PCA of Salmon-derived...”).
Figure 1 caption: This might be the first time that I suggest reducing the text in the caption. You are repeating a lot of the details from your methods.
Lines 153ff: Since your two components together do not surpass 50%, I would suggest moving Figure 2 into the supplementary material and just writing that two components were not sufficient to explain the variance in the data.
Figure 4: Right now, reading the graphs is challenging. Please consider placing increasing the size of each graph and creating a 1x4 presentation, rather than a 2x2. Also consider using a shared legend, as the colour scheme used in each graph is the same.
Lines 218ff: A lot of this also reads like a method. Please move them into the method section instead (up until around line 245).
Line 269 ff: Same as previous comment (up until line 291).
Table 1: Consider moving this into the supplementary material. You do not discuss each in turn, and although the results are relevant, they do not need to be in the main text.
Section 2.5: Start this at line 350, the first lines lend themselves to the methods or discussion.
Line 359f: This reads close to a discussion. I would consider moving this to the discussion, or re-writing it to not interpret the results.
Line 378: Please consider creating dose-response curves for select expression, as you highlight a pattern.
Line 422ff: There are very few refences in this section of 3.2. I appreciate that for a novel exposure regime, there will be limited papers to refer to, but at the moment this section reads like a data interpretation, not a discussion. Please consider re-working this to allow embedding in existing research.
Section 3.3.: Same comment as above.
Line 486: Replace capital N with n please.
Section 4.1. : Please include licencing information and housing details (water temperature, light cycles, breeding group set-up, and water conditions) or refer to a paper that includes these details for your laboratory.
Lines 754ff: I am missing information on the number of embryos exposed per treatment group, quality control for egg collection, etc.
Line 753: What do you mean with “5 samples”? x individuals for 5 replicates? Or were X animals from various replicates pooled to create 4 samples.
Line 754: Please write how animals were euthanised prior to lysing. I assume cold shock in liquid nitrogen?
Reviewer 2 Report
Comments and Suggestions for Authors
North et al. analyzed the effects of the selective serotonin reuptake inhibitor (SSRI) citalopram on zebrafish embryonic development. Their extensive RNA-seq analysis across embryos exposed to different concentrations provides valuable insights in this field. The study is logically presented and, in my opinion, merits publication following minor revisions.
The discussion section is somewhat repetitive and overly long. The number of cited references is also quite high. I encourage the authors to streamline this section to improve clarity and readability.
Several transcription factors with significant differences are listed, but gene duplication is common in zebrafish. For example, in the figure, “EGF” could refer to either egfa or egfb, and Hoxc13 exists as Hoxc13a and Hoxc13b. Providing such detailed information is very important for clarity and reproducibility.
The expression “To our knowledge, … first …” is used twice. Since data presented in a manuscript are expected to be novel, I recommend avoiding this phrasing, or at least not repeating it.
